



# US East Coast synthetic aperture radar wind atlas for offshore wind energy

Tobias Ahsbahs[1], Galen Maclaurin[2], Caroline Draxl[2], Christopher Jackson[3], Frank Monaldo[3], Merete Badger[1]

[1] DTU Wind Energy Risø, Roskilde, Denmark

[2] National Renewable Energy Laboratory, Golden, Colorado, USA

[3] Applied Physics Laboratory, Johns Hopkins University, Baltimore, Maryland, USA

Correspondence to: Tobias Ahsbahs ttah@dtu.dk

## Abstract

We present the first synthetic aperture radar (SAR)-based offshore wind atlas of the US East Coast from Georgia to the Canadian border. Images from Radarsat-1, Envisat, Sentinel-1A, and Sentinel-1B are processed to wind maps using the Geophysical Model Function (GMF) CMOD5.N. Extensive comparisons with 6,008 collocated buoy observations revealed that biases of the individual system range from -0.8 to 0.6 m/s. Unbiased wind retrievals are crucial for producing an accurate wind atlas and intercalibration for correcting these biases by adjusting the normalized radar cross section is applied. The intercalibrated SAR observations show biases in the range of to -0.2 to 0.0 m/s, while at the same time improving the root mean squared error from 1.67 to 1.46 m/s. These intercalibrated SAR observations are, for the first time, aggregated to create a wind atlas. Monthly averages are used to correct artefacts from seasonal biases. The SAR wind atlas is used as a reference to study wind resources derived from the Weather Research and Forecasting (WRF) model. Comparisons focus on the spatial variation of wind resources and show that model results estimate lower coastal wind speed gradients than those from SAR. At sites designated for offshore wind development by the Bureau of Ocean Energy Management, mean wind speeds typically vary between 0.3 and 0.5 m/s for SAR and less than 0.2 m/s for the WRF model within each site. Findings indicate that wind speed gradients and variation might be underestimated in mesoscale model outputs along US East Coast.

## 1    Introduction

Offshore wind energy has been established on the continental shelf of Northern Europe since 2001 with a total installed capacity of 15,780 MW (Wind Europe, 2018). The US East Coast is similar in water depths and population density and could thus be well-suited for offshore wind farms (Kempton et al., 2007). During the past decade, the Bureau of Offshore Energy Management (BOEM) has leased out areas designated for offshore wind farm development along the US East Coast



(BOEM, 2018), and the first wind plant became operational in 2016 (Block Island Wind Farm, Rhode Island[1]). Accurate and long-term wind statistics across broad geographic extents (i.e., wind atlases) are needed to support offshore wind energy deployment. Wind atlases can be developed from local in situ measurements, i.e., buoys or meteorological masts (Troen and Petersen, 1989); numerical weather prediction models; (Dvorak et al., 2013; Hahmann et al., 2015); or satellite-based remote

sensing (Christiansen et al., 2006; Hasager et al., 2015). The objective of this study is to create and validate a satellite-based offshore wind atlas for the US East Coast and compare it to results from numerical weather prediction models.

Offshore wind resource data for the US East Coast are available from the Weather Research and Forecasting (WRF) model (Draxl et al., 2015b; Dvorak et al., 2013). For offshore wind energy, locations close to shore are most attractive because installation costs increase with the distance to shore because of increased water depth and longer cables. The BOEM lease

areas are mainly located in coastal waters less than 70 km from shore where influences from upstream land masses are still substantial (Barthelmie et al., 2007) and mesoscale models can result in high uncertainties (Hahmann et al., 2015). These models need validation. Colle et al. (2016) point out that observations at turbine hub heights around 100 m are lacking and provide case-study-based validation using observations from airplanes. Long-term reference wind climates at broad geospatial scales are missing because observations from ocean buoys are sparse. Radar images from satellites provide

measurements over large areas and it is possible to infer wind speeds from the radar backscatter of the ocean surface. Scatterometers and synthetic aperture radar (SAR) can measure wind speeds following this principle at scales around several hundred kilometers. SAR is better suited for resolving wind resources in coastal zones because of its higher resolution (Christiansen et al., 2006). It has been shown that SAR-derived winds can accurately depict wind speed gradients outwards from 1 km offshore (Ahsbahs et al., 2017) and that SAR-derived show similar mean wind speed variations as experienced by

a wind turbines (Ahsbahs et al., 2018).

Wind resources can be assessed from SAR (Christiansen et al., 2006) and studies have been performed at different locations (Doubrawa et al., 2015; Hasager et al., 2011). For the US East Coast, a SAR-based wind atlas has been created from Radarsat-1 (RS1) data for a small area off the coast of Delaware (Monaldo et al., 2014). Expanding this study to the entire US East Coast with RS1 data is not possible because the images were acquired specifically for this region and coverage

outside this region is limited. We have acquired additional data from Envisat (ENV), Sentinel-1A (S1A), and Sentinel-1B (S1B) that are distributed via Copernicus services. These data are openly available to public research, which is not the case for data from other missions such as TerraSAR-X, Cosmo SkyMed, or Radarsat-2.

The objective of this article is to produce and validate a SAR observation-based wind atlas for the US East Coast by merging four different satellites. We will remove possible offsets between wind retrievals from different SAR sensors and validate

this by comparing with data from the well-established ocean buoy network on the US East Coast. For comparison, we will use the Wind Integration National Dataset (WIND) toolkit (WTK) produced by the National Renewable Energy Laboratory (NREL) from 7 model years of WRF outputs (Draxl et al., 2015b). We focus on coastal wind speed gradients and determine

---

[1] http://dwwind.com/project/block-island-wind-farm/




how they are represented in wind atlases from SAR and WTK. Lastly, mean wind speed variation is examined within BOEM lease areas designated for wind farm development.

The article is structured as follows: Section 2 provides an overview of the data and the area of interest of this study. Section 3 describes the methods used to create a SAR-based wind atlas. Section 4 presents wind climatologies and measurement artefacts of the SAR wind atlas. Section 5 focuses on using the SAR wind atlas to investigate wind variations and compares to the WTK. Sections 6 and 7 further discusses the results and draws conclusions on the potential use of the wind atlas.

## 2 Data and area of interest

### 2.1 Area of interest

We focus this study on coastal waters off the US East Coast from Georgia to the Canadian border. The area of interest is defined as between 30.7° and 45° latitude and -63° and -81.3° longitude extending out 400 km offshore, as shown in Figure 1. The positions of buoys described in Section 2.3 are shown as well.

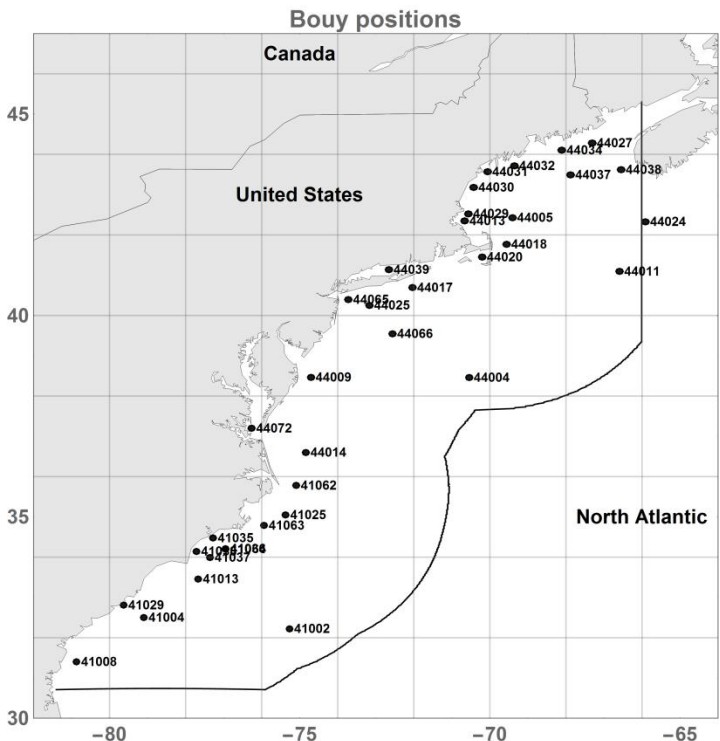

Figure 1: Area of interest for this study and buoy positions.

### 2.2 Synthetic aperture radar

Satellites carrying SAR instruments have been operational for decades and extensive archives exist. Portions of these archives can be used by the scientific community and European Space Agency (ESA) archives are becoming increasingly





open via Copernicus[2] services. SAR sensors usually operate in different modes depending on the desired resolution of the images. We focus on modes that offer the widest possible swaths, because the aim is to create broad-scale wind resource maps. Co-polarized images in VV and HH mode from Envisat's wide swath mode (WSM), Sentinel-1's extra wide (EW) and interferometric wide (IW) modes, and Radarsat-1's ScanSAR wide (WD1) mode are used throughout this study (Table 1).

The number of scenes can be misleading when assessing the coverage of each sensor, because the length of the swath varies (Envisat scenes tend to be more than ten times longer than Sentinel-1 scenes). Sentinel-1A and B are operational at the moment and data to May 2018 are included.

**Table 1: Overview of SAR sensors and the respective imaging modes and properties. The period of operation and number of scenes included in this study are also shown.**

| Satellite | Mode | Polarization | Incidence | Swath width | Period | Scenes |
|---|---|---|---|---|---|---|
| Envisat | WSM | VV | 18–45 | 405 | 2002–2012 | 2198 |
| | | HH | 18–45 | 405 | | 513 |
| Sentinel-1A | IW | VV | 30–45 | 250 | 2015–2018 | 2403 |
| | EW | HH | 30–45 | 400 | | 27 |
| Sentinel-1B | IW | VV | 30–45 | 250 | 2015– 2018 | 517 |
| Radarsat-1 | WD1 | HH | 20–45 | 400 | 1996–2008 | 924 |

**2.3    Buoy data**

High-quality wind and temperature measurements are available on the US East Coast from the buoy center of the National Oceanic and Atmospheric Administration (NOAA) (National Data Buoy Center, 1971). These will be used as reference measurements. We only use buoys more than 5 km from the shore to avoid possible land contaminations in the SAR images. Measurements from 31 buoys are used in this study and the approximate locations are shown in Figure 1. Buoys are mainly

located within 100 km from the shore. Wind speeds and direction are measured every hour for 8 minutes and data are automatically quality controlled (National Data Buoy Center, 2009). We performed additional quality control by checking time periods where SAR wind retrievals showed more than 10 m/s difference. Four periods are removed from specific buoys that showed unrealistically low wind speeds in the buoy measurements for several months and one short period where the buoy measurements are unreasonably high. Locations and measurement heights are recorded annually in the buoy file but

changes can occur within a year. Additional metadata on buoy position and measurement heights are available that represent the most accurate information according to the NOAA buoy center (National Data Buoy Center, 2015). The more accurate metadata have been used.

---

[2] https://www.copernicus.eu/de



## 2.4    WIND toolkit

The WTK was originally developed to support the next generation of wind integration studies with input from experts at NREL in production cost modelling and atmospheric science. The WRF model version 3.4.1 was used to create the meteorological data set, using ERA-Interim reanalysis data as inputs. The meteorological data set has a spatial resolution of

2x2 km and 5 min temporal resolution. It covers 7 years (2007–2013) and is available over the contiguous 48 US states, including the outer continental shelf. The WTK has been used by various research centers within NREL and by universities in multiple studies. A validation report is available for six onshore sites and three offshore sites (Draxl et al., 2015a).

## 3    Methods

### 3.1    Synthetic aperture radar wind retrievals

Level-1 SAR data are downloaded from the data providers and calibration is applied to obtain the radar backscatter measured as the normalized radar cross section (NRCS). The processing is done using the SAR Ocean Products System (SAROPS) software package (Monaldo et al., 2014). Radar backscatter on the ocean surface is determined by Bragg scattering (Valenzuela, 1978) and the NRCS of the ocean surface can be linked to a characteristic wind speed using a geophysical model function (GMF). For C-band radars, the C-band Model (CMOD)-family of functions is most widely used and

CMOD5.N is used for this study (Hersbach, 2010). The resulting wind speed is the equivalent neutral wind at 10 m above the ocean. CMOD5.N is tuned for co-polarized vertical (VV) SAR observations and an incidence angle dependent polarization ratio is applied before processing co-polarized horizontal (HH) images (Mouche et al., 2005). For SAR wind retrievals, the wind direction needs to be known a priori. Wind directions are taken from global weather models from 10 m wind vectors and are interpolated spatially to match the SAR images. Wind direction inputs for the SAR wind retrieval are

obtained from  the National Center for Atmospheric Research Climate Forecast System Reanalysis (CFSR) reanalysis data are used until 2010 and Global Forecast System (GFS) data from 2011 onward.

Radarsat-1 was one of the early operational SAR systems and some of the scenes are known to have problematic distortions (Vachon et al., 1999); i.e., when stitching the subswaths together or with correct geolocation. These typically cause overestimated NRCS values and thus wind speeds that are too high. Because these problems are easy to detect visually but

hard to formalize, Radarsat-1 data have been visually checked and problematic scenes are excluded. Additionally, NRCS values above 44° incidence angle are removed because of frequent unrealistically high NRCS values.

### 3.2    Merging synthetic aperture radar wind fields from different sensors

SAR-derived wind speeds should correctly represent the wind conditions compared to in situ observations, but validation of SAR-derived wind speeds routinely leads to biases that are not consistent between studies (Christiansen et al., 2006;

Horstmann et al., 2002; Lu et al., 2018; Takeyama et al., 2013). Deviations between studies can partially be explained by different GMFs that are used, but also by inconsistent calibration of NRCS values. Biases in the SAR-derived wind speeds




are problematic because they translate to biases in the derived wind atlas. It is particularly problematic to have offsets in the biases between sensors, because these will introduce variability where spatial coverage of sensors changes over the study area. To date, SAR wind atlases have used a singular sensor or – if multiple sensors were merged – inherent differences have not been taken into account (Hasager et al., 2015; Karagali et al., 2018).

Badger et al. (2019) have found systematic differences in the bias when comparing wind speeds retrieved from Envisat and Sentinel-1A/B against in situ observations. Biases for Envisat showed a strong incidence angle dependency and the bias drift over the sensor's lifetime. Badger et al. (2019) found that these biases can be corrected. NRCS are calculated from modelled wind speeds and compared to the SAR measurements. A linear fit of the NRCS differences depending on the incidence angle is then subtracted from the SAR images before retrieving the wind speeds. We apply the reported correction factors for

Envisat and Sentinel-1A/B, which also account for the initial calibration problems of Sentinel-1A before 2015-11-25 (Miranda, 2015). Sentinel-1A data are split in two time periods, before calibration (BC) and after calibration (AC). Corrections for Radarsat-1 are not available in Badger et al. (2019) and, therefore, we calculate adjustment factors from the available Radarsat-1 data using the same methodology. In accordance with recommendations from Badger et al. (2019), Envisat data below 20° have been excluded from the analysis because of increased scatter and bias in the adjustment method.

### 3.2.1 Synthetic aperture radar – buoy comparisons

Comparisons between SAR and buoy measurements are conducted to confirm if results found in Badger et al. (2019) are consistently present in this data set from the United States and whether the suggested adjustment method can remove biases between the sensors. Images during three strong wind storms and SAR wind speeds exceeding 30 m/s have been removed from the comparison because co-polarized SAR wind retrievals are expected to perform poorly in these conditions.

Comparisons between wind speed from buoys and SAR need to account for inherent differences in the measurements. SAR images are matched with the closest buoy time stamp with a maximum difference of 30 minutes (Monaldo, 1988). SAR winds are instantaneous, and they are averaged spatially to a 3 km by 3 km cell to better match the temporal average of buoy measurements. Anemometers on buoys are typically mounted between 3 and 5 m while SAR winds are tuned to 10 m. Buoys wind speeds are therefore extrapolated to 10 m equivalent neutral wind using the Coupled Ocean–Atmosphere Experiment

COARE 3.0 algorithm with temperature measurements from the buoys (Fairall et al., 2003).

Figure 2 shows comparisons between SAR and buoys as scatter plots for SAR winds processed as described in Section 3.1. Comparisons for all collocations in (a) show a slight bias for SAR to overestimate wind speeds by 0.30 m/s. The RMSE of 1.67 m/s is within the targets for satellite wind accuracies of 2 m/s (Figa-Saldaña et al., 2002). Distinguishing between sensors show that biases vary. Large biases towards overestimation of 0.62 and 0.82 m/s are respectively found in Envisat

(b) and Sentinel-1A BC (e), while Radarsat-1 (c) is underestimating wind speeds by 0.89 m/s. Both Sentinel-1A AC (f) and Sentinel-1B (d) have neglectable biases. The results for Envisat and the two Sentinels are in line with observations in Badger et al. (2019).





**Figure 2: Scatter plots of SAR versus buoy winds at 10 m with default processing for: (a) all data, (b) Envisat, (c) Radarsat-1, (d) Sentinel-1 BC, (e) Sentinel-1AC, and (f) Sentinel-1B.**



Figure 3 shows comparisons after applying the intercalibration process described in Section 3.2. Comparisons for data from all satellites have improved both in terms of bias from 0.30 to -0.04 m/s and RMSE from 1.67 to 1.46 m/s. Considering each of the sensors separately, biases lie between -0.2 and 0.03 m/s, which is a drastic improvement compared to biases in Figure 2 ranging between -0.89 to 0.82 m/s. Large improvements are found for Envisat, Radarsat-1, and Sentinel-1 BC both in terms of biases and RMSE. Closely examining the two largest data sets Envisat (b) and Sentinel-1A AC (e), an overestimation of less than 7 m/s and an underestimation of more than 9 m/s can be observed. These opposing biases are averaged to nearly zero bias. The intercalibrated SAR winds have smaller biases than the individual data sets and small differences between the sensors compared to the default processing. The following analysis will therefore be based on these adjusted SAR wind maps.



**Figure 3: Scatter plots of SAR versus buoy winds at 10 m with incidence angle adjusted processing for: (a) all data, (b) Envisat, (c) Radarsat-1, (d) Sentinel-1 BC, (e) Sentinel-1AC, and (f) Sentinel-1B.**





### 3.3 SAR wind atlas methods

A wind atlas is a map of statistical representations of the wind speed over a designated area. The wind climate is typically represented by a Weibull distribution of wind observations that is characterized by the Weibull scale parameter (A [m/s]) and

the shape parameter (k [unitless]). They are related to the mean energy density (E [W/m^2]) by:

$$E = \frac{1}{2}\rho A^3 \Gamma \left(1 + \frac{3}{k}\right) \text{(1)}$$

where ρ is the air density, and Γ the Gamma function. The mean wind speed can be defined as the arithmetic mean of the available samples.

$$U = \frac{1}{F} \sum_n u_n \text{(2)}$$

With the wind speed of the individual image $u_n$ and the total number of observations F.

A typical approach in wind energy is to use the Wind Atlas Analysis and Application (WAsP) program that implements methods from the first European wind atlas (Troen and Petersen, 1989). Wind atlases are generated by taking the mean wind speed from long time series, but it is also possible to use the quasi-instantaneous wind fields derived from SAR (Christiansen

et al., 2006). Properties of the acquisition such as temporally fixed overpasses, relatively low sampling, and data truncation lead to uncertainty in a SAR-based wind atlas (Barthelmie and Pryor, 2003). A special version of WAsP developed for satellite-based wind atlases (S-WAsP) is used in this study. Weibull fitting uses 2nd moments as recommended in Pryor et al. (2003). SAR wind scenes are averaged on a regular WGS84 grid with 0.02° cell spacing before processing the data to a wind atlas. Results from a SAR-based wind atlas can be noisy because of the high resolution of wind fields and the relatively

few samples. Therefore, we apply a Gaussian filter using a standard deviation of 0.03° with a cutoff at 0.06° to smooth the mean wind fields.

Sensors acquire images at fixed times of day, which will can cause a bias in the mean wind speed where diurnal cycles are present. Advanced methods for classifying SAR wind maps are available (Badger et al., 2010). We chose to apply random sampling considering all images available, which is recommended where more than 400 images are available (Pryor et al.,

2003). This approach is also used in earlier SAR-based wind atlases (Hasager et al., 2011). The influence of diurnal cycles is investigated in more detail in Section 4.1.1.

SAR scenes are acquired primarily for purposes other than wind resource assessment. This can influence the temporal coverage of acquisitions. One example is sea ice detection that will mainly occur during the winter months (Sandven et al., 1999). The study domain is located in the mid-latitudes and winds are expected to change with the seasons. We therefore

check for seasonal biases in the data acquisition, described in Section 4.1.2. For the arithmetic mean wind speed in Eq. 2, the seasonal bias can be corrected by calculating mean wind speeds $U_m$ by month (Monaldo, 2011):

$$U_m = \frac{1}{F_m} \sum_n u_{n,m} \text{(3)}$$





with the number of observations $F_m$ for each month $m$ and $u_{n,m}$ the SAR wind speeds occurring in this month. Monthly mean wind speeds are then averaged to a seasonally corrected mean wind speed $U_{sc}$:

$$U_{sc} = \sum_m \frac{F_m}{F} U_m \quad (4)$$

S-WAsP is not able to account for seasonal biases in the Weibull parameter estimation. Therefore, no seasonally corrected

Weibull parameters or energy densities are available.

## 4   Results

### 4.1   Wind resource statistics

In the following, we present the first SAR-based wind atlas for the US East Coast based on intercalibrated SAR wind fields from four systems. Figure 4 shows wind statistics at 10 m: (a) The arithmetic mean wind speed from Eq. 2 calculated from

SAR and (b) the mean wind speed from modelled data (WTK). A visual comparison shows similar features. WTK wind speed contours are smoother than those from SAR for two reasons: i) SAR-derived wind speeds are based on high resolution observation that can resolve sub-kilometer scale variation in the wind fields, and ii) SAR-derived mean winds are derived from fewer samples while WTK are based on 7 full years of hourly modelled wind speeds. Wind speeds are lower close to the coast and increase with the distance from shore. A band of high winds is located off the coast of North Carolina and

extends to the northwest with higher mean wind speeds in SAR than in the WTK. Horizontal variations of the SAR wind speed are higher than for WTK. The mean wind speeds are lower for SAR than WTK in a region close to the shores of Virginia and Delaware. Another clear difference is the wind speed in the Gulf of Maine. WTK data show winds of less than 7.5 m/s while SAR winds go up to 8.5 m/s. In both data sets, a feature of lower mean wind speed is present to the southeast of Nantucket but more pronounced in the SAR-derived map.







**Figure 4: Wind atlas maps at 10 m for: (a) arithmetic mean SAR wind speed, (b) mean wind speed from WIND toolkit, (c) number of SAR samples, (d) SAR energy density; (e) SAR Weibull scale parameter A; (f) SAR Weibull shape parameter.**





North of 34° latitude, more than 350 samples are used, but fewer than 250 are used off the coast of Georgia (Fig. 4c). The energy density ranges from 200 W/m$^2$ close to shore to 800 W/m$^2$ far offshore (Fig. 4d). The Weibull shape parameter A (Figure 4e) shows similar features as the wind speeds and the energy density. The scale parameter k (Figure 4e) ranges between 2 and 3 in the south and 1.75 and 2.5 in the north of the domain. High k values are associated with a narrow Weibull

5    wind speed distribution.

Wind resources and wind roses are compared between SAR, WTK, and in situ buoy measurements for three example locations along the coast in Figure 5. Buoy 44029 is located in the Gulf of Maine, buoy 44009 is located off the Delaware coast, and buoy 41038 is located off North Carolina; see Figure 1 for detailed positions. Buoy data are filtered to cover full years (at least 80% available data) to avoid seasonal sampling biases; between 7 and 10 years of measurements are available

10   at the buoy locations. WTK covers 7 full years and the SAR winds are sampled over the entire period from 1998 to 2018 but less frequently. SAR wind speeds are expressed as equivalent neutral wind while the 10 m wind speeds from WTK are stability-dependent wind speeds. Buoy wind speeds are extrapolated accordingly but stability effects are small (less than 0.2 m/s differences for the mean wind speed).



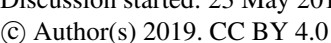



**Figure 5: Weibull fits and wind roses for buoys 44029 (a-c), 44009 (d-f), and 41038 (g-i). SAR on the left (a,d,g), WIND toolkit in**

5 **the middle (b,e,h), and wind roses for buoy, SAR, and WIND toolkit on the right (c,f,i). Key characteristics are given in the tables:**

**Number of observations (N), Weibull shape (A), Weibull scale (E). Units of the tables are: [N] = --, [A] = m/s, [k] = --, [E] = W/m².**



SAR-based results show good agreement at buoy 44009 but distributions are skewed towards higher wind speeds at 44029 and lower at 41038, while WTK distributions generally agree well with the buoy data. Wind directions for SAR show more winds from the northwest for buoy 44029 and agree well with buoy data otherwise. Wind directions from the WTK show most deviations for buoy 41038. There are large deviations ranging from -136 to 72 W/m$^2$ between wind resources as measured from buoys and SAR that merit closer investigation in the following.

### 4.1.1 Diurnal cycle

As noted in Section 3.3, SAR sensors acquire data at fixed times. We investigate influences this might have on the wind retrieval by investigating the diurnal cycle at five buoys located across the study domain. In addition to the three buoys in Figure 5, buoy 44065 is located south of New York City and 44072 is in Chesapeake Bay. All buoys are within 20 km of shore with the exception of buoy 44009, which is located approximately 60 km offshore. The shape of the diurnal variation is similar between these buoys with minima between 10 am and 12 pm local time; see Figure 6. Buoy 44009 shows less diurnal variation, which might be connected to this buoy being located further offshore. Overlaid in Fig. 6 is a histogram of the satellite acquisition times showing that scenes are not randomly sampled as expected from the polar orbit. We note that the afternoon times of ENV overpass coincide with a minimum in the diurnal variation, while S1A, S1B, and RS1 overpasses are outside this time interval.

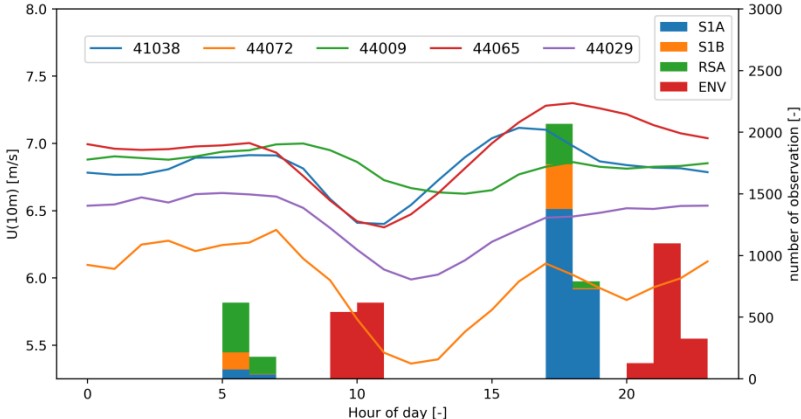

**Figure 6: Mean wind speed of selected buoys in local time (UTC – 5). The bars indicate the number of satellite observations for Radarsat (RSA), Envisat (ENV), Sentinel-1A (S1A), and Sentinel-1B (S1B).**

### 4.1.2 Seasonal sampling bias

We investigate seasonal sampling biases in SAR for four regions of 2° by 2° along the coast. Figure 7 shows the spatial average of monthly acquisition frequency $F_m/F$ and monthly mean wind speed $U_m$ from Eq. 3 and 4 at four areas. Acquisitions are unevenly distributed over the year. More data are available in the winter in the Gulf of Maine while



Delaware and North Carolina are biased towards late summer to early autumn. $U_m$ shows considerable seasonal changes with generally lower winds in summer and higher winds in winter.

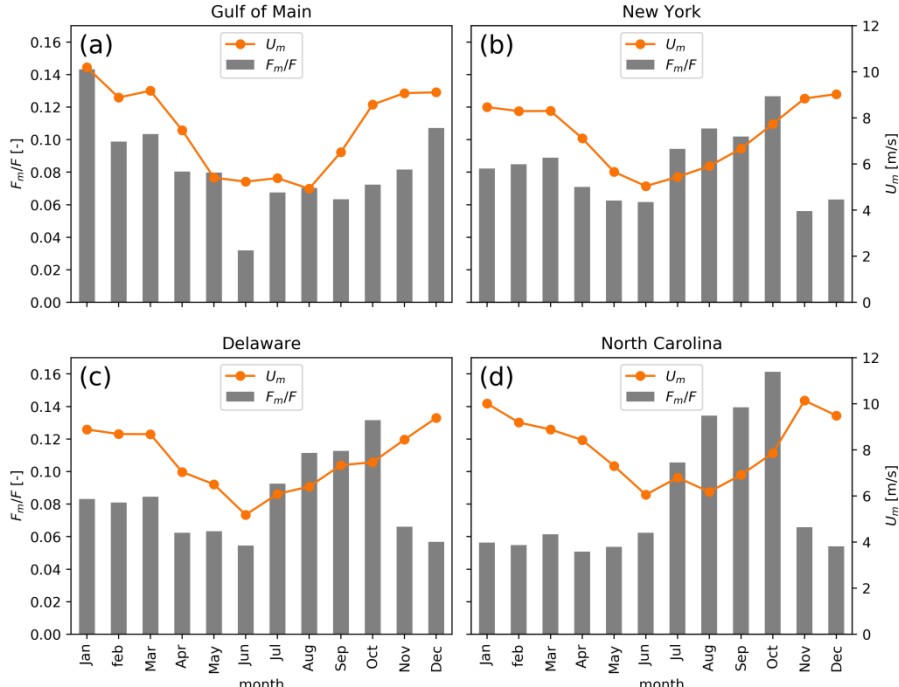

**Figure 7: Frequency of data acquisition (Fm/F) and mean SAR wind speeds (Um) averaged by month over four regions close to: (a) Gulf of Maine, (b) New York, (c) Delaware, (d) North Carolina**

Figure 7 shows considerable seasonal sampling biases. A seasonally corrected SAR (SAR_SC) mean wind speed map is calculated from Eq. 4 and shown in Figure 8a together with the differences to uncorrected maps from Fig. 8b. Seasonal correction reduces wind speeds in the north, while it increases wind speeds in the south of the study domain.



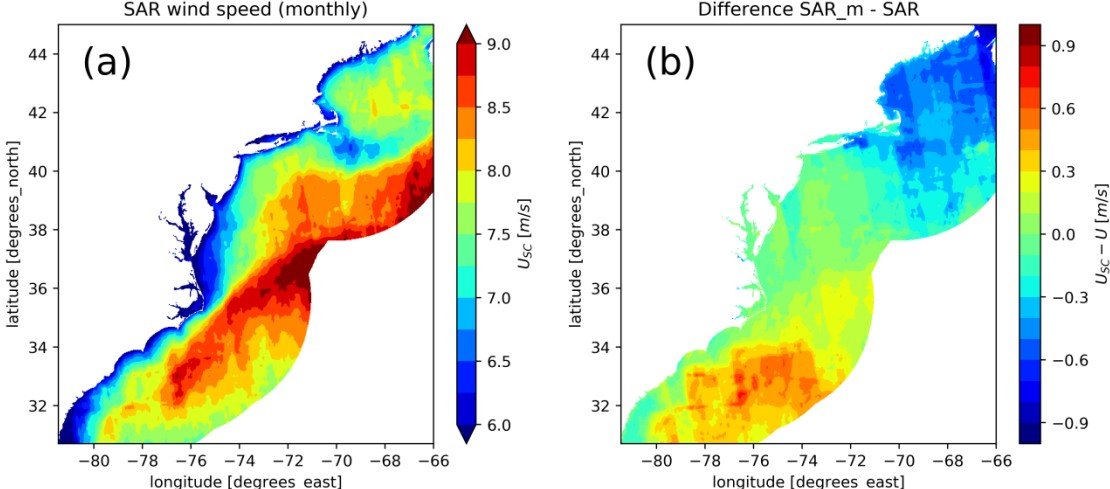

**Figure 8: (a) Seasonally corrected SAR wind speed (b) difference between seasonally corrected and original SAR map.**

Two SAR-based mean wind maps in Figure 4a and Figure 8a have been calculated. The one better representing the long-term wind conditions is determined from comparison to long-term mean wind speeds from ocean buoys. Buoys are required

to have at least 7 full years (more than 80% recovery rate) of measurements. It is necessary to use a representative position for the buoys because buoy positions can change over time and SAR or WTK are not collocated in time. This requires that buoy positions do not change significantly during the measurement period. Buoy 41002 and 44018 are removed because their location changes more than 100 km. Sixteen buoys fulfil these criteria and statistics on comparisons to the SAR mean wind speed in Figure 4a, WTK mean wind speed in Figure 4b, and the seasonally corrected SAR mean wind speed in Figure

8 are presented in Table 1.

**Table 2: Mean absolute error (MAE), root mean square error (RMSE), and bias between buoys as well as SAR (U), seasonally corrected SAR ($U_{SC}$), and WTK.**

|        | MEA  | RMSE | Bias |
|--------|------|------|------|
| SAR    | 0.51 | 0.63 | 0.34 |
| SAR_SC | 0.3  | 0.39 | 0.09 |
| WTK    | 0.24 | 0.30 | 0.15 |

The seasonally corrected mean wind speed $U_{SC}$ shows lower RMSE, MAE, and bias. We consider this to be a better

representation of the seasonality. $U_{SC}$ will therefore be used for comparisons with the WTK in Section 4.2.



## 4.2    Spatial wind variability from SAR and WTK

To compare SAR and WTK mean wind speeds in the coastal zone, we define transects perpendicular to the generalized coastline of the United States up to 100 km from shore. Because of the complexity of the shoreline, a compromise needs to be found between perpendicular transects, avoiding crossing transects, and the definition of distance to shore for convex corners. The resulting transects are shown in Figure 9 and are labelled with unique identifiers (transect_id) ranging from 0 in the north to 650 in the south.

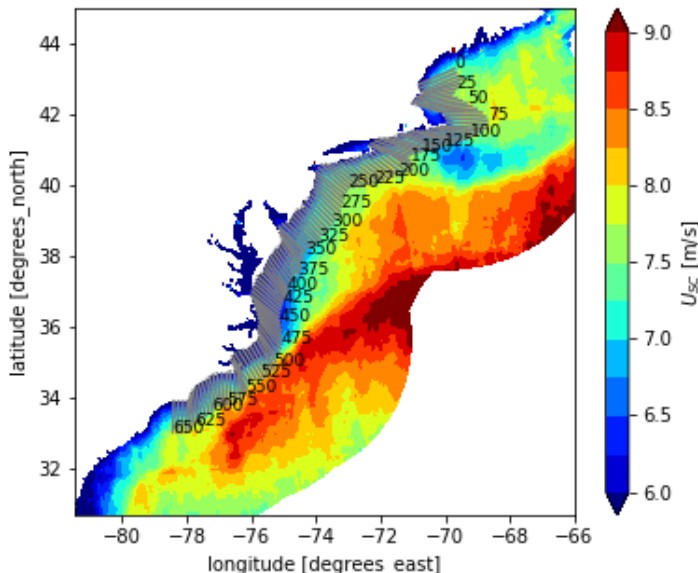

**Figure 9: Seasonally corrected SAR mean wind speed maps with transects perpendicular to shore (every fifth transect is plotted). Starting at transect_id 0 in the Gulf of Maine going to transect_id 650 off North Carolina.**

Wind speeds are linearly interpolated along each transect every 2 km. Figure 10 shows the wind speeds per transect ID and as a function of distance to shore. These plots can be seen as a horizontal sheet of mean wind speeds along the coastline perpendicular and parallel to shore. The white areas are land contamination in SAR wind maps originating from islands not accounted for in the generalized coastline. Again, we can see similarities in the features on large scales with a band of high wind speed between transect_id 500 and 600 but also smaller features like an increased wind speed at the mouth of the Delaware River around transect_id 350.





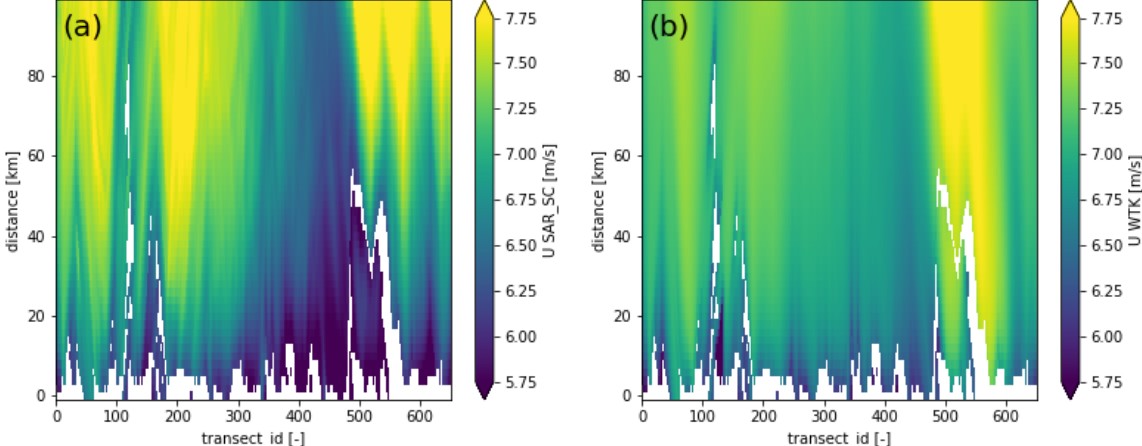

**Figure 10: Wind speeds at 10 m over all transects as a function of distance to shore. (a) SAR wind speed seasonally corrected, (b) WIND toolkit.**

The presentation of wind speeds in Figure 10 represents spatial structures of the mean winds along the coast but it is hard to assess differences visually. We will focus on wind speed variations in two directions: along-shore and perpendicular to the coastline. The latter is commonly referred to as a coastal wind speed gradient.

### 4.2.1 Along-shore variation

Figure 11 shows wind speed transects along the shore averaged over distances to shore of [10,20], [20,30], [40,60], and [60,100] km. From transect_id 0 to 300 the two transects closest to shore (Fig. 11 a and b) show remarkably good agreement, both absolute and in shape. Further offshore, in panels (c) and (d), the positions of local maxima and minima are similar but the amplitude of these features is larger for SAR than WRF. From transect_id 300 onward (southward), SAR gives consistently lower wind speeds with the exception in panel (d) around transect_id 570 and 650.





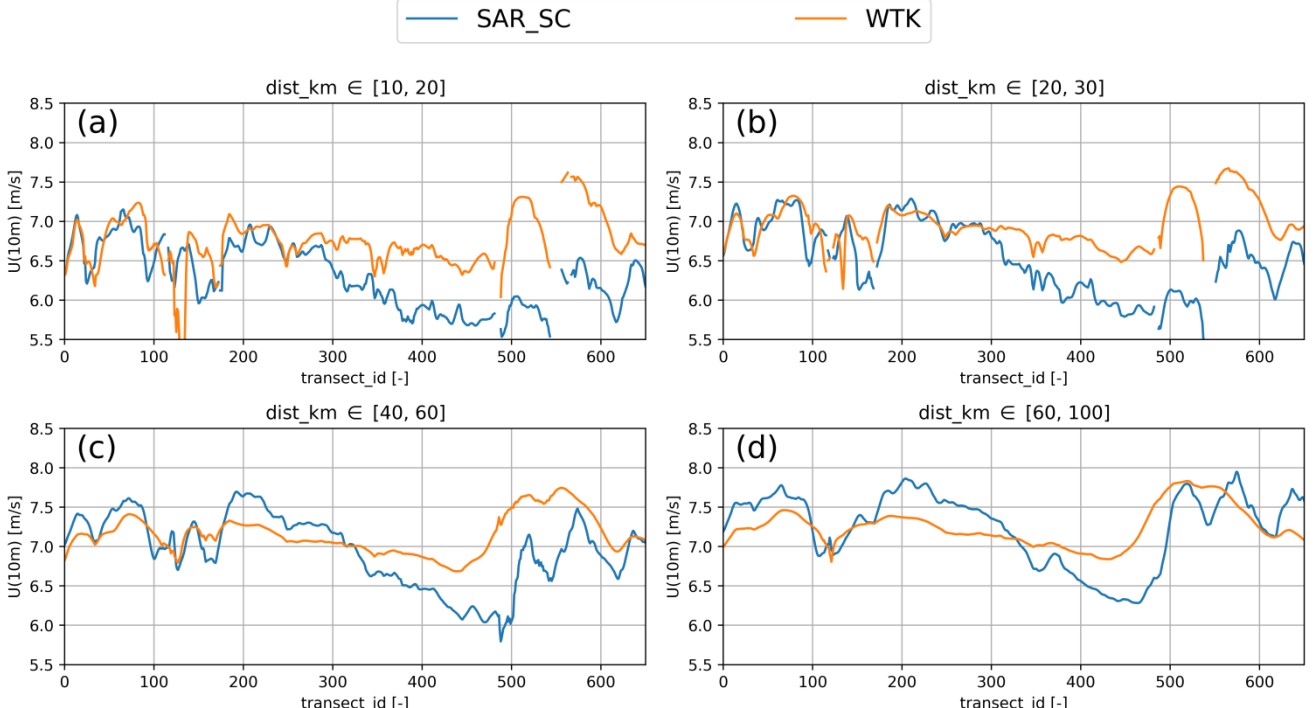

**Figure 11: Along-shore variation of the mean wind speeds for distance to shore intervals of: (a) [10,20], (b) [20,30], (c) [40,60], and (d) [60,100] km.**

The region closer than 60 km to shore is most interesting for wind farm development. Here, SAR observations suggest high wind speeds in the north up to transect_id 250, while WTK consistently shows higher wind from transect_id 500 southward.

### 4.2.2 Coastal gradients

Wind speeds averaged over the distance to shore for six regions are shown in Figure 12. All regions show coastal gradients with the typical increase in mean wind speed with distance from shore. For Fig. 12a, around Nantucket, there is very good agreement both in the gradient and in the absolute value. Regions (b), (c), and (f) show similar behavior, with SAR exhibiting lower wind speeds closer to shore but higher gradients resulting in higher SAR winds further offshore. Gradients are similar for (d) but SAR winds are offset by 0.7 m/s toward lower wind speeds. The most pronounced differences in terms of wind speed gradients are found around Pamlico Sound (e) with SAR winds up to 1.5 m/s lower close to shore and a steep gradient from 40 to 100 km offshore.





**Figure 12: Mean wind speeds averaged over several transects covering six different regions. From north to south (a) Nantucket, (b) Long Island, (c) State of New York, (d) Virginia to Delaware, (e) Pamlico Sound, (f) southern part of North Carolina.**

The transects show consistently higher wind speed gradients with the exception of the most northern region around Nantucket (Figure 12a). The wind speed gradient is defined as:

$$grad = \frac{dU}{dx}$$

where $x$ is the distance to shore. The wind speed gradient is averaged for each transect resulting in 650 mean gradients for SAR and WTK. A distribution thesis is shown in Figure 13. Mean gradients are mostly positive, indicating higher wind speeds further offshore as expected. For WTK, the distribution is almost symmetric with a mean of 0.91 m/s per 100 km.




The distribution from SAR is more skewed and clearly separated from the WTK. The mean of the distribution is 1.40 m/s per 100 km, which is considerably higher than the WTK .

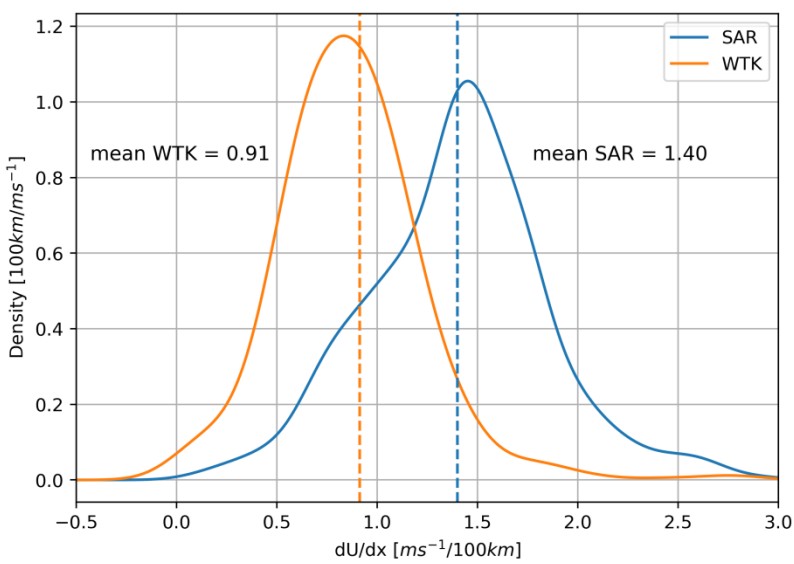

**Figure 13: Density plot of wind speed gradients from SAR and WIND toolkit. Dashed lines indicate the mean.**

### 4.2.3    Wind resource variation within Bureau of Offshore Energy Management areas

Wind farm development is allowed within the limits of the offshore lease areas defined by the BOEM; see Figure 14. Because lease areas are typically several hundred square kilometers large, wind resources are expected to vary within each of the areas. Information on the magnitude of this variation is needed for wind farm development.





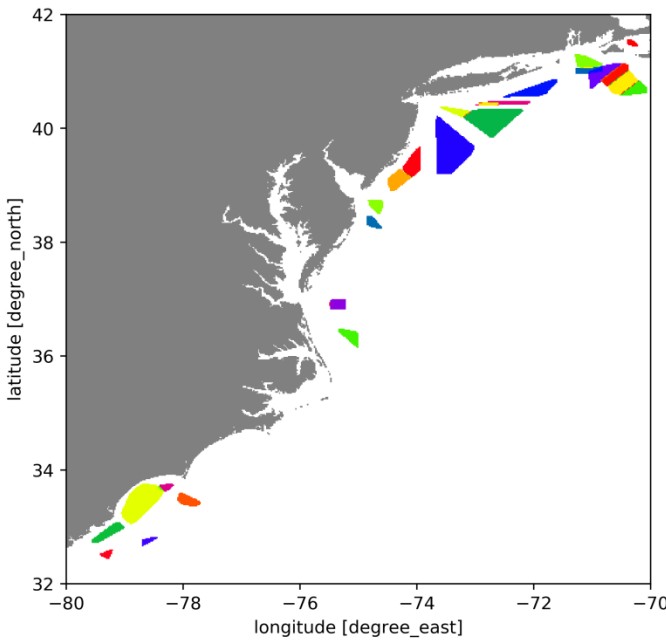

**Figure 14: Bureau of Offshore Energy Management lease areas for the US East Coast. Color codes are used to differentiate different areas.**

We select mean wind speeds from SAR and the WTK at all grid points within a lease area. The distribution of mean wind
speeds within each area is then calculated and presented as a violin plot in Figure 15. The variation of mean wind speeds is
higher from SAR for all areas except Cape Wind and Kitty Hawk. The average of the differences between minimum and
maximum are 0.2 m/s for the WTK and 0.47 m/s from SAR. This indicates that the WTK predicts much less variation of
wind resources within a potential wind farm site than SAR. Note that a mesoscale numerical weather prediction model such
as WRF is unable to pick up wind speed variations in the order of 0.5 m/s, and their RMSEs are typically more than 0.5 m/s.





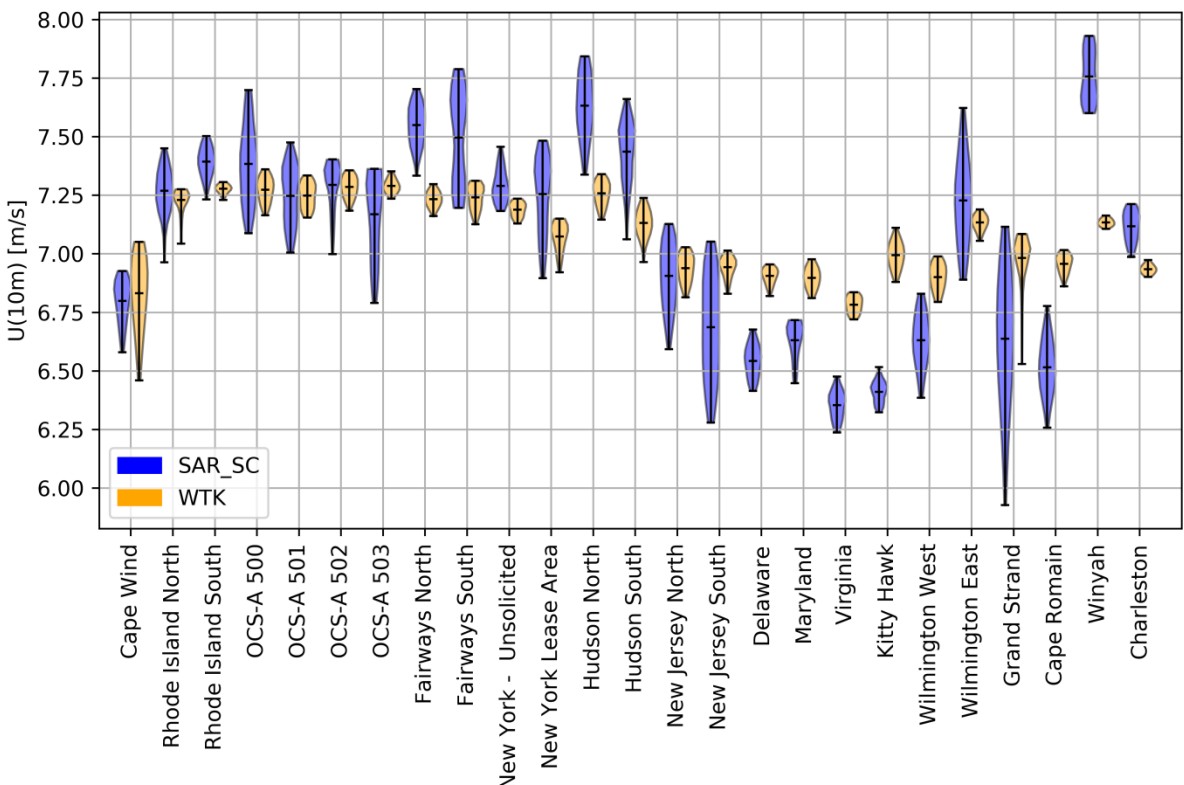

**Figure 15: Violin plot for wind speeds from seasonally corrected SAR and WIND toolkit. Potential lease sites are ordered from north (left) to south (right).**

## 5    Discussion

In the following, we discuss the results from the evaluation of the SAR wind atlas and the associated artefacts from the sampling. The representation of wind speed variations in the coastal zone is discussed as measured from SAR and modelled in the WTK.

a)    Validation of intercalibrated SAR wind archive with buoys

This study presents the first SAR wind atlas merging archives from four sensors into a consistent data set. Extensive comparisons with buoys show that even though data are processed consistently with CMOD5.N, biases between the sensors range from -0.89 to 0.62 m/s (see Figure 1), which is similar to results from northern Europe. Badger et al. (2019) suggested intercalibration to remove biases by adjusting the NRCS as a linear function of the incidence angle using modelled wind speed and direction inputs. Those adjustments decrease the difference in biases between sensors to 0.2 m/s. Overall, a



tendency to overestimation for low wind speeds and underestimation for high wind speeds remains in the SAR wind, which influences the Weibull fitting. Two findings speak to the generality of this approach: i) Intercalibration tables derived over northern Europe can be applied for the US East Coast; ii) applying the suggested intercalibration method to Radarsat-1 data reduces the bias to 0.03 m/s. This concept should in no way substitute efforts to better understand scattering mechanisms that

improve GMFs (Troitskaya et al., 2018). Tuning NRCS values is an application-driven approach to produce wind maps with consistently low bias. This is in contrast to previous and current efforts to determine the most suitable GMF for SAR wind retrievals (Christiansen et al., 2006; Takeyama et al., 2013).

   b)   SAR wind atlas for the US East Coast

We have produced a SAR-based wind atlas of the US East Coast covering the coast from Georgia to the Canadian border. An alternative to SAR measurements are scatterometers (Stoffelen and Anderson, 1997). From a wind resource perspective, their main advantage is the higher temporal resolution, but it comes at the cost of a lower spatial resolution of typically 25 km. Merging SAR and scatterometer wind data to create a wind atlas has been done (Hasager et al., 2015), but this approach needs further refinement to fully utilize the high temporal coverage from scatterometers and the high spatial resolution from

SAR.
The Weibull parameters A and k, energy density, and mean wind speeds are calculated from the available SAR wind maps; see Figure 4. The energy density, Weibull shape A, and mean wind speed generally increase with the distance from shore. The Weibull scale parameter k is high in the south and lower in the north. The Weibull k parameter requires more samples than wind speed or shape parameter to be correctly estimated (Barthelmie and Pryor, 2003). The area with high shape

parameters coincides with a low number of samples and significant seasonal bias in the sampling, which casts doubt on the accuracy of results in these instances.
The presented SAR wind atlas is calculated at a 10 m height; however, for wind energy applications, estimates closer to turbine hub heights would be more desirable. Extrapolation of the results is possible using model-derived stability inputs (Badger et al., 2016). For the purpose of comparing the wind atlas to the WTK, it is desirable to have the SAR data as

independent from modelling results as possible. Extrapolating these results is therefore beyond the scope of this study but would increase the applicability of the SAR-derived wind speeds for wind energy purposes  and will be considered in the future. Comparisons at 10 m are nonetheless valuable to assess differences relative to mesoscale models and to gain insight in the horizontal variation of wind resources.

c)   Wind resource comparisons

This study shows a comparison between SAR and WTK, assuming that both are representative for the wind climatology. A method to sample SAR winds according to wind classes based on modelled climatology is available to reduce the number of samples necessary for wind resource assessment (Badger et al., 2010). This has the advantage of choosing scenes that are





more representative for the large-scale wind conditions but comes with the disadvantage of weakening the independence of the SAR-derived results. Combining data with other in situ observations as done in Doubrawa et al. (2015) is an option, but the density of measurements on the US East Coast is lower than in the Great Lakes, where the method was developed.

SAR-derived wind resources overestimate the energy density by 72 W/m$^2$ at buoy 44029 and underestimate them by 136

W/m$^2$ for buoy 41038. Sampling of the SAR scenes shows considerably uneven sampling between different seasons (Figure 7). In the region of buoy 44029, the winter months with high wind speeds are overrepresented. Wind resource estimates derived from these data will retain a bias to higher wind speeds and thus overestimate the wind resource, which is in line with our observations. For buoy 41038, an opposing bias towards summer and early autumn associated with low wind speeds could explain the underestimation of wind resources. The resource estimate from SAR shows little difference in Weibull

parameters and the energy density for buoy 44009. In this region, seasonal sampling is more evenly distributed and oversampling occurs between the extrema. Wind resource estimates from the WTK have been made at the same buoy locations. Generally, the wind resources are estimated more accurately than from SAR but overestimations of 69 W/m$^2$ occur at buoy 44029.

SAR wind atlases for other regions have generally not reported seasonal dependency in the data coverage (Hasager et al.,

2011; Karagali et al., 2014). A simple method to overcome this problem was implemented using weighted monthly averages to calculate the mean wind speed (Eq. 4) (Monaldo, 2011). This seasonally corrected mean reduces wind speeds in the north, while increasing them in the south of the study domain. Differences frequently exceed 0.5 m/s, which are substantial for a product that should be used in the context of wind energy. The seasonally corrected mean wind agrees better with long-term means from buoy observation (Table 2), both in terms of mean errors and RMSE, and we see them to be the better choice

when estimating a wind climatology. Using monthly weights in the estimation of SAR-derived Weibull parameters should be possible but implementing and validating such a method is beyond the scope of this article.

d)   Influences of diurnal variability

SAR satellites operate on orbits with fixed times for ascending and descending tracks 12 hours apart. This sampling bias influences results in the wind atlas. The time of day of the observations from Envisat are approximately 10 am and 10 pm, while Sentinel-1A, 1B, and Radarsat-1 are observing at 5 am and 5 pm. Envisat contributes the most observations in this study and thus its temporal bias will largely influence results. For buoy 44072 located in the Chesapeake Bay, both Envisat acquisition times are close to a local minimum of the wind speed. Therefore, a bias towards underestimating the

climatological mean wind speed is expected here. For SAR mean winds at the remaining buoy locations displayed in Figure 6, the effect from diurnal variability will be smaller but still present. Adding more Sentinel-1 acquisitions will even out the diurnal sampling bias from Envisat.

Sea breeze phenomena present in this region are contributing to diurnal wind speed variations (Hughes and Veron, 2015). The influence of the diurnal cycle is more pronounced closer to shore. SAR images that happen to oversample the wind





speed minimum of the diurnal cycle would cause a stronger bias towards lower wind speeds closer to shore than farther offshore. Wind observations sampled in such a way would artificially increase the coastal gradient.

e) Wind speed gradients from  and the WIND toolkit

Mean wind speed maps from the WTK are compared with seasonally corrected SAR winds. Mesoscale models are known to have higher uncertainties offshore if winds come from land (Hahmann et al., 2015). For the buoy locations in Figure 5, westerly winds come across land and the wind roses show that these directions occur frequently. For these directions, coastal wind speed gradients are expected to occur, caused by the roughness change between land and sea. Coastal gradients from

SAR have been shown to agree well with lidar wind speeds for the first few kilometers from shore (Ahsbahs et al., 2017). The SAR-based wind atlas has a resolution of 0.02° that makes it ideal for investigating horizontal variations in the mean wind speed and serving as a reference for modelled wind speed gradients from the WTK.

Wind speeds from SAR are typically lower than those from the WTK close to shore but gradients from SAR are higher than from the WTK for most regions (Figure 12b, c, e, and f). 100 km from shore SAR tends to give higher winds than WTK.

Wind speed gradients show that SAR winds are, on average, showing an increase of 1.40 m/s per 100 km. For the WTK, this value is only 0.91 m/s per 100 km. Fixed times of the satellite tracks could influence wind speed gradients if they show diurnal variability. This cannot easily be investigated from buoy measurements because they lack the spatial coverage, which was the initial motivation for this study. The influence of sampling biases is unlikely the sole source for the observed differences in the wind speed gradients. It seems likely that WTK is underestimating wind speed gradients for the first 100

km but a closer investigation is necessary to confirm this.

Wind atlases can be used to investigate wind resource potential at large scales; i.e., identifying regions that are most promising for wind farm development. Mean wind speeds at equal distance to shore showed good agreement in the northern part of the domain but disagree more strongly in the south. Results from SAR show much more variation along the coast and a distinct minimum in the wind speed close to Delaware (transect_id from 400 to 500). In this region, results from buoy

44072 located at Chesapeake Bay (Figure 7) suggest that Envisat is sampling during two local wind speed minima. This could partially explain the low SAR winds speed in this region. Results from earlier resource assessments in Dvorak et al. (2013) using WRF found that wind resources are generally increasing going further from south to north in our investigated domain but show less variability than both SAR and WTK.

Spatial variation of the mean wind speed within lease areas for wind farm development is investigated using WTK and SAR

data (Figure 15). For most areas, WTK shows less variation than SAR. For example, mean wind speeds from the WTK for "New Jersey South" range from 6.8 to 7.0 m/s. Low variation like this might lead a developer to neglect horizontal wind speed gradients at their site; i.e., during the planning of a measurement campaign. At the same location, SAR wind speeds range from 6.3 to 7.1 m/s. This variation is substantially larger, suggesting that wind speed variation within this area should be considered. SAR wind maps resolve more variation than mesoscale models or scatterometers, which can explain part of





the increased variation (Karagali et al., 2014). Another reason could be speckle noise in the SAR images themselves but spatial and temporal averaging, as performed in this study, will greatly reduce this effect. Variations found here are in line with previous studies from the Anholt wind farm in Denmark, which is located downstream of a complex coastline and can experience strong wind speed gradients (Ahsbahs et al., 2018; Peña et al., 2017).

f) Future work

We see two ways forward for using SAR-derived wind atlases for mesoscale model comparisons: i) improving the climatological representativeness of the SAR wind atlas, and ii) dropping the assumption of random sampling of the SAR
data and subsample the mesoscale model data to match the sampling characteristics of SAR.

i) Seasonal biases and poor representation of diurnal cycles are likely major contributors to uncertainty for the SAR wind atlas. We would like to suggest a way to further improve SAR wind atlases to correctly represent climatological conditions. Weighting SAR scenes by month could overcome seasonal biases and give better estimations of the Weibull parameters while retaining the observational character of the SAR-based wind atlas. Additionally, acquisition times of Envisat and
Sentinel-1 are separated by 5 hours as shown in Figure 7. With an increasing archive of Sentinel-1 data, future wind atlases will be based on data more evenly distributed over the time of day.

ii) The WTK is based on a mesoscale model and does not experience sampling biases because it provides time series data for each point. Instead of assuming the SAR wind atlas to be climatologically representative, we suggest randomly sampling mesoscale model data to more realistically represent the seasons and times of day present in the SAR archive. Repeating this
process would create an ensemble of model-based wind atlases including uncertainties from the SAR sampling. The SAR wind atlas can raise awareness for possible flaws in the model where it falls outside the mesoscale model ensemble envelope. This approach could be combined with the investigation of spatial wind speed variability presented in this study.

## 6    Conclusion

Using a large number of collocated buoy measurements, we have shown that SAR wind fields from different sensors can be
intercalibrated. The derived SAR wind atlas is novel in two regards: 1) it ensures consistent calibration towards wind retrievals from different sensors, and 2) it covers the US East Coast where a similar product has not been available before. The presented sensors show seasonal sampling biases that are inconsistent over the study domain but mean wind speeds can be corrected to show a bias of 0.09 m/s and an RMSE of 0.39 m/s compared to long-term buoy observations.

Comparisons of the long-term mean wind speeds at 10 m between SAR and WTK indicate that: 1) the model could under-
predict the horizontal wind speed gradient with respect to the distance to shore, and 2) wind speed variations within areas designated for offshore wind farm development are lower in the WTK than with SAR. These findings raise awareness that





spatial variations of wind resources might be underestimated from in this mesoscale model. SAR-derived wind atlases can serve as independent data sources most useful in the early planning phase of an offshore wind farm project.

**Acknowldgements**

This work was authored [in part] by the National Renewable Energy Laboratory, operated by Alliance for Sustainable Energy, LLC, for the U.S. Department of Energy (DOE) under Contract No. DE-AC36-08GO28308. Funding provided by the U.S. Department of Energy Office of Energy Efficiency and Renewable Energy Wind Energy Technologies Office. The views expressed in the article do not necessarily represent the views of the DOE or the U.S. Government. The U.S. Government retains and the publisher, by accepting the article for publication, acknowledges that the U.S. Government retains a nonexclusive, paid-up, irrevocable, worldwide license to publish or reproduce the published form of this work, or allow others to do so, for U.S. Government purposes. We thank Evan Rosenlieb, Michael Rossol and Paul Doubrawa from the National Renewable Energy Laboratory (NREL) for their input and assistance. We also acknowledge Jane Lockshin from NREL, who created the offshore transects used to examine coastal gradients in wind speed estimates and George Scott who assisted with handling ocean buoy data download from NOAA.

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
