# Peer review of "US East Coast synthetic aperture radar wind atlas for offshore wind energy"

_Wind Energy Science, 2019_

## Short Comment (SC1) · 23 May 2019

Dear authors, many thanks for this paper, which I read with great interest. The detailed explanation of the SAR data processing is very useful, and I learned many interesting things. I have made some comments as I read the paper, you will find them below. I hope that at least some of them will be relevant for your work.

All the best, Rémi Gandoin. C2Wind.

—-Section 1—-

Page 1 Line 25: "on the continental shelf of Northern Europe since 2001". The first offshore wind farm was commissioned in the Baltic Sea Denmark in 1991, do you mean the "North Sea Continental Shelf" ?

[Figure]

Page 2 Line 2: "long-term wind statistics". Consider adding "at hub height".

Page 2 Line 16: "at scales around several hundred kilometers". You mean the swath width is o(100km) ? Or the length scales of the wind patterns ?

Page 2 Line 17: "SAR is better suited for resolving wind resources in coastal zones because of its higher resolution". Here, I would like to ask: 1) is the SAR resolution mentioned here the "raw" resolution, or the 10m neutral wind speed spatial resolution, after the GMF has been applied ? 2) How does this resolution compare with the WRF resolution (i guess you mean the actual physical resolution of the scales that WRF can resolve, that is up down to but not smaller than what is left to the PBL schemes, and not the grid size))? 3) From 1) and 2), why is that increased resolution a benefit ?

Page 2 Lines 18 and 19 : "accurately" and "similar". Can you quantify ?

Page 2 Line 20. Can you here mention as well the limitations of SAR in shallow waters ?

—Section 2.1—

Page 3 Line 11: "The positions of buoys". I may be wrong, but at this stage I have not seen a reference to the NDBC database (where the data originate I guess). Can you refer to it ?

Figure 1 : Can you add water depths contour lines, as well as wind farms lease areas ? Aren't you missing 44008, the Nantucket Shoals buoy ?

—Section 2.20—

Table 1: please add units to the table headers.

—Section 2.3—

Page 4 Line 11: "High-quality wind and temperature measurements". How do you define "high quality" and argue for it ?

Page 4 Line 17: "from specific buoys". Which ones ?

—Section 3.1.—

Page 5 Line 15: "equivalent". What does "equivalent" means in this context ?

Page 5 Line 18: "needs to be known a priori". With what accuracy and precision ? What is the wind direction is wrong by 5,10, 30° ?

Page 5 Line 19: "to match the SAR images". What does this mean ?

Page 5 Lines 20 and 21: Please check the wording.

–Section 3.2—

Page 5 Line 28: "should". Please explain. As I understand fron the information presented until now, these could only be compared reasonably with spatially averaged (to some, yet unspecified, spatial resolution) measured wind speed values at 10m in neutral conditions (can these be found ? I remember the buoys are about 4 m tall, and the atmosphere rarely neutral).

General remark: (Badger et al, 2019) is quite a central piece here, yet the paper is not yet available (and not yet submitted as I understand). This is limiting the interpretation of the results, as I am curious to understand what are these biases, how they have been quantified and how they can be corrected. Could you maybe disclose at least some of the main results and bias correction figures ?

—Section 3.2.1—

Page 6 Line 19: "are expected to perform poorly in these conditions." Reference missing, possibly https://ieeexplore.ieee.org/document/8016397 ?

Page 6 Lines 21-22: "SAR winds are instantaneous". I remember reading that SAR images were approximately 1-minute averages, what do you mean by "instantenous" ?

Page 6 Line 12: "matched with the closest buoy time stamp": does the validation

improve by interpolating between the 8-minute hourly measurements ?

Page 6 Lines 22-23. "better match the temporal average of buoy measurements". Can you explain how 3 km was chosen ?

Figure 2: Can you show density scatter plots instead of regular scatter plots ? Can you compute for every buoy the mean bias and the RMSE, and display these values on a histogram ? It could be interesting to show these in a Table and on a map as well. Depending on the amount of samples in each stability class (stable/unstable), the same information information could be presented for both of these subsets (for each buoy).

—Section 4.1—

Figure 4: Could these maps be shown for the periods Oct-Mar and Apr-Sep ? As explained in https://findit.dtu.dk/en/catalog/2443553221, stability conditions greatly vary along the East Coast, due to the Gulf Stream, the Labrador current, the polar vortex and the Azores high. In Southern New England for instance, cold water in shallow waters (like Georges Bank) lead to stable conditions during summer, and thereby small wind speeds at the surface. I note that the large wind speed area in (a) correlate well with the outer continetal shelve bathymetry contour line (see https://www.researchgate.net/profile/Jason_Link/publication/228680188/figure/fig3/AS:667854709993487@15362403797 of-the-Northeast-US-continental-shelf-ecosystem-which-is-inhabited-by-pollock-Depth_W640.jpg), and thereby it could be interesting to understand whether the map could be biased by differences in stability conditions. Could you show the 100mMSL wind speed from WTK ?

–Section 4.1.2—

Table 2: please add units to the table headers.

—Section 4.2.1—

Figure 11: Section no. 300 marks a clear difference between the SAR
and WTK data close to the coast. How can this be explained ? Maybe by colder water temperatures in reality than in the WTK model ? See cross sections of SSTs from MODIS for the year 2018 on this Figure: https://www.dropbox.com/s/0r26vzovcfw1eoj/SST_transects_large.png?dl=0.

—Section 4.2.3—

Page 23 s 8-9: "Note that a mesoscale numerical weather prediction model such as WRF is unable to pick up wind speed variations in the order of 0.5 m/s, and their RMSEs are typically more than 0.5 m/s.". Please provide specific references.

—Section 5—

Pages 24 Line 1: "tendency to overestimation for low wind speeds and underestimation for high wind speeds". Is this more than a tendency, maybe a consistent negative bias in unstable conditions and a positive bias in stable conditions ? Could you quantify "overestimation", "underestimation", "low" and "high" wind speeds ?

Page 25 Line 27-28: " Comparisons at 10 m are nonetheless valuable to assess differences relative to mesoscale models and to gain insight in the horizontal variation of wind resources.". I suggest to add "provided that the stability conditions are the same across the area of interest".

---

## Short Comment (SC2) · 28 May 2019

Dear Remi,

thank you for your comments. I have gone through them and agree that answering them will clarify the manuscript. I will wait with a detailed response until the referee comments arrive, as coordination with all co-authors takes time.

I can give some general answers: NDBC buoy 44008: We used an XML file published by the NDBC to choose relavant buoys and some of the "moored buoys" are incorrectly labeled. We corrected some, but missed 44008.

Badger et.al. (2019): The paper was submitted already in November 2018 and we hope that the review process would be done by now, but it took much longer than

expected. We are in the review process now and will resubmit this soon. I will post the paper at once, when it is accepted to give access to the pre-print.

Best regards,

Tobias Ahsbahs

---

## Referee Comment (RC1) · Anonymous Referee #1 · 9 Jul 2019

**Review of "US East Coast synthetic aperture radar wind tlas for offshore wind energy" by Ahsbahs et al, submitted to Wind Energy Science Discussions**

The paper provides the first atlas of the offshore wind resource along the US East Coast derived from high spatial resolution (2 km) SAR products from four satellites at 10 m above sea level. A detailed comparison against two independent data source is also provided, buoys and WIND Toolkit, as well as a discussion of biases, seasonality, and gradients. This is one of the best papers I have ever reviewed and it should definitely be published soon. I only have a few minor comments and suggestions.

1. It was not until page 5 (line 15) that the 10-m height was mentioned. It is important to let the reader know that the atlas is valid at 10 m asl earlier than that. I recommend that you add this information in the abstract ("We present the first synthetic aperture radar (SAR)-based, offshore, 10-m wind atlas …") and even in the Introduction around p. 2 lines 20-30.
2. Similarly, you need to mention the height of the buoy measurements in section 2.3 (it is mentioned later (p. 6 line24) but it should be here) and the height of the WIND Toolkit output in section 2.4 (10 m – is this a real model level or an interpolated value? If interpolated as I think, how?)
3. In the abstract, the WRF model is mentioned, but also the WIND Toolkit project should be mentioned, otherwise the reader thinks that the authors ran the WRF model themselves. Instead, they used WIND, which is a well-documented, publicly available dataset.
4. P. 2 , l. 19: a noun is missing "SAR-derived [what?] show…"
5. P. 4, l. 5: what is a "scene"? A snapshot? A picture?
6. P. 5, l. 13: what exactly is the "normalized radar cross section of the ocean surface"? I am unclear on what exactly it is that the SAR measures on the ocean surface. Is it a reflectivity of some sort? Is it related to white caps of the waves? Please add a brief description here. Also, briefly describe how the GMF works.
7. P. 5, l. 20. Either a period "." is missing (before "Climate Forecast…") or the phrase is incomplete. It's good that it was not the WIND's wind direction that was used here.
8. P. 6, l. 25: briefly describe how the extrapolation from 5 m to 10 m is calculated in the COARE 3.0 algorithm for the buoy data.
9. Figs. 2 and 3: what are the black lines with vertical error bars? Medians? Please add info in the captions.
10. Fig. 4: please use the same color bar for Fig. 4a and 4b. They are similar but not identical in the current figure.
11. Fig. 7: the month of "Feb" should be capital.
12. P. 27, l. 4: missing noun after "from", maybe "SAR"?
13. Future work, item ii). I do not agree with this recommendation, remove it or explain it better. Why would randomly sampling model output, which actually includes seasonal and diurnal variability correctly, be a better way to present a wind atlas? This procedure would mimic the SAR behavior, but it would not necessarily provide a better estimate of the actual wind resource. I think the authors are saying that this random sampling

method would be a better to validate SAR, not a better way to represent the wind resource. If so, please clarify/rephrase.

---

## Short Comment (SC3) · 11 Jul 2019

Dear reviewer,

thank you for taking the time to reading our manuscript and provide your comments. We are also glad to hear that you liked it.

We are waiting for the second reviewer to comment and will start updating the manuscript taking all comments provided into account.

Best regards, Tobias Ahsbahs

———————————————————

---

## Short Comment (SC4) · 20 Mar 2020

SUMMARY

This is a valuable manuscript demonstrating data quality control and analysis methods that should be of considerable use to the offshore wind energy community. The manuscript provides a well described application of methods from prior studies to the offshore waters of the United States' East Coast. The downside of this approach is that the current study retains all of the meteorological and statistical shortcomings of these existing methods. The advanced data sources, particularly WRF model analyses, used in this study, provide the authors with a, so far, unexploited opportunity to correct these short comings and set a new standard for SAR wind power analysis. My comments

below focus on highlighting these opportunities.

STRENGTHS

- Good choice of geophysical model function

OPPORTUNITIES

Opportunity 1 - Neutral vs stratified surface layer.

A longstanding challenge in SAR wind analysis has been that neutral stratification of the surface layer "must" be assumed. This has resulted in SAR retrieval algorithms returning estimates of neutral-equivalent wind rather than of true wind. The resulting neutral-equivalent wind is actually a proxy for surface stress, just expressed as wind via the neutral drag law.

The effect of this assumed neutral stratification of the surface layer is a wind speed bias that depends on the stability of the atmospheric surface layer. The SAR-derived wind speeds are too low in regions where the surface layer is stable, because wind speed must compensate for the too high (i.e. neutral rather than stable) drag coefficient assumed. Likewise, the SAR-derived wind speeds are too high in regions where the surface layer is unstable, because wind speed must compensate for the too low (i.e. neutral rather than unstable) drag coefficient assumed. Basically, SAR-derived wind is having to compensate for the lack of the stability dependence of the vertical mixing of momentum in the surface layer. This is reflected in the present study in the observation that SAR winds are faster than buoy winds over the Gulf Stream (where the atmospheric surface layer is destabilized by the warm underlying water) and slower than the buoy winds over the cold waters north of the Gulf Stream (where the atmospheric surface layer is stabilized by the cool underlying water).

For most of the history of SAR, that was the best anyone could do, because there were no good sources for surface layer stability estimates over the ocean. This study, however, has the access to WRF analyses from which surface layer stability can be

easily calculated. In Section 3.2.1 - The TOGA COARE bulk flux algorithm is used to account for the effects stability on the vertical extrapolation of buoy winds. This same stability correction could be used to convert SAR-derived surface stress to stability-aware SAR-derived winds. All it would take would be to use the neutral drag law to convert the neutral-equivalent SAR-derived winds to surface stress and then the equations from the TOGA COARE bulk flux algorithm to convert that surface stress back to a stability-aware 10 m wind. This would be a major advance for SAR wind analysis, one the authors are perfectly positioned to make given that they are already using both WRF analyses (from which surface layer stability can be calculated) and the TOGA COARE bulk flux algorithm which allows their affects on the flux/wind relationship to be computed.

Locations where this issue comes up include: Page 2 lines 14-15 Page 5, line 15 Section 3.2.1 - all Page 11, lines 15-16 Page 13, Line 12 Page 17, Figure 8 - The Gulf Stream's northwest edge is so prominent in this figure precisely because of the lack of stability correction in the neutral-equivalent SAR-derived winds. Page 18, Figure 9 - Same. Page 20, lines 12-14 - This is another sign that the change in surface layer stability across the northwest edge of the Gulf Stream is contributing to the gradient in neutral-equivalent SAR-derived winds observed there. Page 21, line 4 - This is due to the cross-talk between surface layer stability and neutral-equivalent SAR-derived winds. Page 25, lines 22-24 - Here is where you basically outline the method I'm suggesting above. In short, you're most of the way there already, so you might as well make the advance and claim the glory.

Opportunity 2 - Weighting cases in Weibull fitting

The authors wisely weight cases to equalize monthly contributions to the mean, but forebear from doing so when fitting the Weibull distribution parameters. I was curious if this latter process was as hard as the authors assumed, so I looked up how Weibull distributions are fit and discovered that weighting data from different months differently in finding the parameters of a Weibull distribution should be straightforward.

See the link below for a clear discussion of how the method of moments is used to find the Weibull parameters. http://www.real-statistics.com/distribution-fitting/method-of-moments/method-of-moments-weibull/Since the inputs to this method are just mean and standard deviation, both of which can be computed with weighted observations, Weibull distributions can be fit with weighted observations with very little coding effort.

Publishing this trivial, but currently unused advance would be of great help to the SAR wind climatology community and would also impact other meteorological communities which are using the method of moments to fit various distributions to data that is unevenly distributed in space or time.

MINOR ITEMS

Page 1, line 22 - "vary" is vague. Some readers will read this sentence as meaning the mean wind speed is under 1 m/s rather than the intended meaning of the mean wind speed varying by this much across a wind-farm lease area. This issue of too general terms being used for statistics for which precise terms or phrases are available recurs in this manuscript. I have attempted to point out each location where reader confusion may arise.

Page 2, line 16 - "at scales around" - This wording will make most readers think the resolution rather than the swath width is several hundred kilometers.

Page 3, line 1 - "variation" is too vague a term. Please specify if you mean temporal or spatial variation and over what time or space scale.

Page 4, Table 1 - I suspect most readers would like a column with SAR pixel size. Also, incidence angle and swath width need units. Degrees and Kilometers, I suspect.

Page 5, Section 2.4 - It is not clear from this paragraph how these pieces fit together. In particular, it should be made clear whether or not WRF part of WTK?

Page 5, lines 19-21 - Please explain why the data source switched.

Page 6, lines 7-8 - "from modeled wind speeds" - It would help readers to know which modeling system you're referring to here.

Page 8, lines 6-8 - What are these numbers and why are they being discussed here. Are they extreme cases? Means? The discussion is to too terse for clarity.

Page 10, paragraph below the second equation - Would it be better to aggregate spatially before fitting the Weibull distribution rather than after? One worries about the order of fitting and smoothing when the fitting is a nonlinear process as it is in this second order moment approach. This is an issue of Jensen's Inequality, I think.

Page 13, line 12 - While the difference is small in the mean, that is in all likelihood because stable cases and unstable cases are roughly equally likely. The stability impact on the tails of the distribution could thus be quite large. The spatial distribution of biases noted by the authors speak strongly to the impact of surface layer stability on the errors in neutral-equivalent SAR-derived winds, even in the mean.

---

## Author Comment (AC1) · 4 Apr 2020

Dear George,

thank you for taking up this review. It has been hanging in the review process since last April. This January I finished my PostDoc and am now working in the private sector which is resulting in a bit longer response time – please excuse this. I will take contact to my co-authors regarding the two opportunities that you very correctly identified and we will see how much of that potential can be add to this paper.

Best regards, Tobias Ahsbahs

---

## Author Comment (AC2) · 6 Jun 2020

**Review of "US East Coast synthetic aperture radar wind tlas for offshore wind energy" by Ahsbahs et al, submitted to Wind Energy Science Discussions**

The paper provides the first atlas of the offshore wind resource along the US East Coast derived from high spatial resolution (2 km) SAR products from four satellites at 10 m above sea level. A detailed comparison against two independent data source is also provided, buoys and WIND Toolkit, as well as a discussion of biases, seasonality, and gradients. This is one of the best papers I have ever reviewed and it should definitely be published soon. I only have a few minor comments and suggestions.

**Answer:** Thank you very much for a fast and very positive review. We highly appreciate your constructive comments and have followed them all as specified in the following.

**1.** It was not until page 5 (line 15) that the 10-m height was mentioned. It is important to let the reader know that the atlas is valid at 10 m asl earlier than that. I recommend that you add this information in the abstract ("We present the first synthetic aperture radar (SAR)-based, offshore, 10-m wind atlas …") and even in the Introduction around p. 2 lines 20-30.

**Answer:** Agreed. We have added this information to the abstract and the introduction.

**2.** Similarly, you need to mention the height of the buoy measurements in section 2.3 (it is mentioned later (p. 6 line24) but it should be here) and the height of the WIND Toolkit output in section 2.4 (10 m – is this a real model level or an interpolated value? If interpolated as I think, how?)

**Answer:** Buoy measurements are at heights between 2 and 7 m above the sea level. We have added sentences at your suggested locations. The 10m wind variables from WRF are used directly.

**3.** In the abstract, the WRF model is mentioned, but also the WIND Toolkit project should be mentioned, otherwise the reader thinks that the authors ran the WRF model themselves. Instead, they used WIND, which is a well-documented, publicly available dataset.

**Answer:** We have added this in the abstract and modified the sentence to: "The SAR wind atlas is used as a reference to study wind resources derived from the Wind Integration National Dataset Toolkit (WTK), which is based on seven years of modelling output from the Weather Research and Forecasting (WRF) model." Throughout the rest of the manuscript, we have used 'WTK' consistently about this data set.

**4.** P. 2 , l. 19: a noun is missing "SAR-derived [what?] show…"

**Answer:** It is SAR derived wind speeds. We have corrected the sentence to:

"It has been shown that SAR-derived winds can accurately depict wind speed gradients measured by ground based lidars near the coastline (Ahsbahs et al., 2017) and that SAR wind fields show similar mean wind speed variations as those experienced by wind turbines (Ahsbahs et al., 2018)."

 **5.** P. 4, l. 5: what is a "scene"? A snapshot? A picture?

**Answer:** In principle, the same scene can be observed from different angles; each leading to their own image. As we are only looking at co-polarized images here, these terms are almost interchangeable. We

have changed all occurrences of "scene" to "SAR image" as this is more understandable for the wind energy community.

**6.** P. 5, l. 13: what exactly is the "normalized radar cross section of the ocean surface"? I am unclear on what exactly it is that the SAR measures on the ocean surface. Is it a reflectivity of some sort? Is it related to white caps of the waves? Please add a brief description here. Also, briefly describe how the GMF works.

**Answer:** The normalized radar cross section is the quantity for the radar backscatter per unit area. GMFs are empirical functions relating the radar backscatter to the wind speed. We have tried to clarify this to the reader:

"SAR wind retrievals from the database of the Technical University of Denmark are used for this study and their processing is described in the following. SAR images are measures of the radar backscatter from the Earth's surface. The intensity of this backscatter is commonly referred to as the normalized radar cross section (NRCS). Level-1 SAR data are downloaded from the data providers and calibration is applied to obtain the NRCS. The processing is done using the SAR Ocean Products System (SAROPS) software package (Monaldo et al., 2014). Radar backscatter and thus the NRCS of the ocean surface is determined by Bragg scattering (Valenzuela, 1978). This scattering mechanism is most sensitive to wave lengths on the order of 10 cm. At this scale, waves can be assumed to be in local equilibrium with the wind speed and therefore, the NRCS and the wind speed are correlated. An empirical Geophysical Model Function (GMF) can link the NRCS and additional radar parameters to the wind speed at 10 m height above the sea surface."

**7.** P. 5, l. 20. Either a period "." is missing (before "Climate Forecast…") or the phrase is incomplete. It's good that it was not the WIND's wind direction that was used here.

**Answer:** We have rephrased to:

"Two sources of wind directions are used for the SAR wind retrieval: until 2010, wind directions come from the National Center for Atmospheric Research Climate Forecast System Reanalysis (CFSR) reanalysis data and from 2011 onwards, wind directions from the Global Forecast System (GFS) are used."

**8.** P. 6, l. 25: briefly describe how the extrapolation from 5 m to 10 m is calculated in the COARE 3.0 algorithm for the buoy data.

**Answer:** We have added the following sentences and a reference to briefly describe this algorithm:

"In this algorithm, atmospheric stratification is described using the difference between the air and sea temperature together with empirically found constants. The wind speed is then extrapolated considering atmospheric stability and roughness as described by Charnock's relation (Charnock, 1955)."

**9.** Figs. 2 and 3: what are the black lines with vertical error bars? Medians? Please add info in the captions.

**Answer:** The black lines indicate the mean SAR wind speed per wind speed bin (binned by buoy winds) and one standard deviation. We have added a description to the text:

"SAR wind speeds are split into 1 m/s bins according to the buoy wind speed. The SAR mean wind speed and standard deviation around this mean are calculated and plotted as well."

And in the caption of the two figures:

"The black curves indicate the mean within each 1 m/s bin and the vertical lines around the mean value indicate one standard deviation within this bin.".

**10.** Fig. 4: please use the same color bar for Fig. 4a and 4b. They are similar but not identical in the current figure.

We changed this. A GIS team NREL is remaking the plots with nicer plotting libraries.

**11.** Fig. 7: the month of "Feb" should be capital.

**Answer:** We have made the change.

**12.** P. 27, l. 4: missing noun after "from", maybe "SAR"?

**Answer:** We have rephrased to "Mean wind speed maps from SAR and WTK have been compared in this study."

**13.** Future work, item ii). I do not agree with this recommendation, remove it or explain it better. Why would randomly sampling model output, which actually includes seasonal and diurnal variability correctly, be a better way to present a wind atlas? This procedure would mimic the SAR behavior, but it would not necessarily provide a better estimate of the actual wind resource. I think the authors are saying that this random sampling method would be a better to validate SAR, not a better way to represent the wind resource. If so, please clarify/rephrase

**Answer:** We agree so we have revised the section on future work completely and removed this point. The focus is now on future perspectives for SAR-based wind atlases alone:

"With an increasing archive of Sentinel-1 data, future wind atlases will be based on samples, which are more distributed over the time of day. The rapid growth of our SAR data archives over time will in itself improve the accuracy of wind resource statistics. Further, a weighting of the SAR scenes by month could partly overcome seasonal biases and give better estimations of the Weibull parameters while retaining the observational character of a SAR-based wind atlas."

---

## Author Comment (AC3) · 6 Jun 2020

**SUMMARY**

This is a valuable manuscript demonstrating data quality control and analysis meth- ods that should be of considerable use to the offshore wind energy community. The manuscript provides a well described application of methods from prior studies to the offshore waters of the United States' East Coast. The downside of this approach is that the current study retains all of the meteorological and statistical shortcomings of these existing methods. The advanced data sources, particularly WRF model analyses, used in this study, provide the authors with a, so far, unexploited opportunity to correct these short comings and set a new standard for SAR wind power analysis. My comments below focus on highlighting these opportunities.

**Answer:** Thanks very much for giving this manuscript a thorough review and for the constructive suggestions for its improvement.

**STRENGTHS**

- Good choice of geophysical model function OPPORTUNITIES

**Answer:** Thanks!

**Opportunity 1 - Neutral vs stratified surface layer.**

A longstanding challenge in SAR wind analysis has been that neutral stratification of the surface layer "must" be assumed. This has resulted in SAR retrieval algorithms returning estimates of neutral-equivalent wind rather than of true wind. The resulting neutral-equivalent wind is actually a proxy for surface stress, just expressed as wind via the neutral drag law.

The effect of this assumed neutral stratification of the surface layer is a wind speed bias that depends on the stability of the atmospheric surface layer. The SAR-derived wind speeds are too low in regions where the surface layer is stable, because wind speed must compensate for the too high (i.e. neutral rather than stable) drag coeffi- cient assumed. Likewise, the SAR-derived wind speeds are too high in regions where the surface layer is unstable, because wind speed must compensate for the too low (i.e. neutral rather than unstable) drag coefficient assumed. Basically, SAR-derived wind is having to compensate for the lack of the stability dependence of the vertical mixing of momentum in the surface layer. This is reflected in the present study in the observation that SAR winds are faster than buoy winds over the Gulf Stream (where the atmospheric surface layer is destabilized by the warm underlying water) and slower than the buoy winds over the cold waters north of the Gulf Stream (where the atmo- spheric surface layer is stabilized by the cool underlying water).

For most of the history of SAR, that was the best anyone could do, because there were no good sources for surface layer stability estimates over the ocean. This study, however, has the access to WRF analyses

from which surface layer stability can be easily calculated. In Section 3.2.1 - The TOGA COARE bulk flux algorithm is used to account for the effects stability on the vertical extrapolation of buoy winds. This same stability correction could be used to convert SAR-derived surface stress to stability- aware SAR-derived winds. All it would take would be to use the neutral drag law to convert the neutral-equivalent SAR-derived winds to surface stress and then the equa- tions from the TOGA COARE bulk flux algorithm to convert that surface stress back to a stability-aware 10 m wind. This would be a major advance for SAR wind analysis, one the authors are perfectly positioned to make given that they are already using both WRF analyses (from which surface layer stability can be calculated) and the TOGA COARE bulk flux algorithm which allows their affects on the flux/wind relationship to be computed.

Locations where this issue comes up include: Page 2 lines 14-15 Page 5, line 15 Section 3.2.1 - all Page 11, lines 15-16 Page 13, Line 12 Page 17, Figure 8 - The Gulf Stream's northwest edge is so prominent in this figure precisely because of the lack of stability correction in the neutral-equivalent SAR-derived winds. Page 18, Figure 9

- Same. Page 20, lines 12-14 - This is another sign that the change in surface layer stability across the northwest edge of the Gulf Stream is contributing to the gradient in neutral-equivalent SAR-derived winds observed there. Page 21, line 4 - This is due to the cross-talk between surface layer stability and neutral-equivalent SAR-derived winds. Page 25, lines 22-24 - Here is where you basically outline the method I'm suggesting above. In short, you're most of the way there already, so you might as well make the advance and claim the glory.

**Answer:** We are grateful to receive this concrete and detailed suggestion for an opportunity, we could take. Although it seems simple to apply air-sea temperature differences in combination with the TOGA COARE algorithm in order to correct the SAR winds for atmospheric stability, we have chosen not to pursue this opportunity in the present manuscript for the following reasons:

- We would like to keep the SAR and WTK data sets completely independent since the main objective of our analyses is to compare the two data sources and explore their strengths and weaknesses in connection with wind resource assessment.

- Previous research indicates that WRF outputs are not so suitable for stability correction of instantaneous wind speed profiles whereas they can be used with confidence for correction of the long-term average wind speed (Pena & Hahmann, 2012, https://onlinelibrary.wiley.com/doi/full/10.1002/we.500; Badger at al., 2016, https://doi.org/10.1175/JAMC-D-15-0197.1).

- The best way to persue the suggested opportunity would, in our oppinion, be to first validate the air and sea temperatures from WTK (WRF) against the ocean buoy observations. If their accuracy is satisfactory, the TOGA COARE algorithm could be applied, as suggested here, and both the uncorrected and corrected wind speeds could then be compared against the buoy observations of wind speed. Altogether, this would be a substantial amount of analyses, which would deserve a separate publication. Given that both the ocean buoy observations and the WTK are open data sets and that the US East Coast remains highly relevant for offshore wind energy developments, we would be very interested in continuing our efforts in the near future.

We have taken the liberty to use paragraphs of text from the reviewer's comments directly in the manuscript in order to properly describe and discuss the issue of atmospheric stability effects. In the discussion:

"A longstanding challenge in SAR wind analysis has been that neutral stratification of the surface layer must be assumed. The effect of this assumed neutral stratification of the surface layer is a wind speed bias that depends on the stability of the atmospheric surface layer. The SAR-derived wind speeds are too low in regions where the surface layer is stable, because wind speed must compensate for the too high (i.e. neutral rather than stable) drag coefficient assumed. Likewise, the SAR-derived wind speeds are too high in regions where the surface layer is unstable, because wind speed must compensate for the too low (i.e. neutral rather than unstable) drag coefficient assumed. Basically, the SAR-derived wind is having to compensate for the lack of the stability dependence of the vertical mixing of momentum in the surface layer. This is reflected in our study in the observation that SAR winds are faster than buoy winds over the Gulf Stream (where the atmospheric surface layer is destabilized by the warm underlying water) and slower than the buoy winds over the cold waters north of the Gulf Stream (where the atmospheric surface layer is stabilized by the cool underlying water). Results from earlier resource assessments in Dvorak et al. (2013) using WRF show that wind resources are generally increasing going from south to north in our investigated domain but show less variability than both SAR and WTK".

And in the section on future work:

"This study has utilized the COARE 3.0 bulk flux algorithm to account for the effects of atmospheric stability on the vertical extrapolation of buoy winds. This same stability correction could be used to convert the SAR-derived surface stress to stability-aware SAR winds given that the air-sea temperature difference for any point in the area of interest can be obtained from the WTK data set. The neutral drag law could be used to convert the neutral-equivalent SAR-derived winds to surface stress and then the equations from the COARE 3.0 bulk flux algorithm could be applied to convert that surface stress back to a stability-aware 10 m wind. This would be a major advance for SAR wind analysis and represents a natural next step for our analysis of wind resources along the US East Coast."

**Opportunity 2 - Weighting cases in Weibull fitting**

The authors wisely weight cases to equalize monthly contributions to the mean, but forebear from doing so when fitting the Weibull distribution parameters. I was curi-ous if this latter process was as hard as the authors assumed, so I looked up how Weibull distributions are fit and discovered that weighting data from different months differently in finding the parameters of a Weibull distribution should be straightforward.

See the link below for a clear discussion of how the method of moments is used to find the Weibull parameters. http://www.real-statistics.com/distribution-fitting/method- of-moments/method-of-moments-weibull/ Since the inputs to this method are just mean and standard deviation, both of which can be computed with weighted observations, Weibull distributions can be fit with weighted observations with very little coding effort.

Publishing this trivial, but currently unused advance would be of great help to the SAR wind climatology community and would also impact other meteorological communities which are using the method of moments to fit various distributions to data that is un- evenly distributed in space or time.

**Answer:** We agree that it is, in principle, not hard to use the method of moments to recalculate the Weibull parameters and the energy densities based on the weighted wind speed values. The issue is more of a practical nature as we have used the DTU-software S-WAsP for the Weibull fitting, which is no longer maintained or updated. The tool does not include functionalities for weighting of SAR wind data. The main advantage of using the S-WAsP tool is its ability to handle large amounts of satellite data and projecting the wind maps on a regular grid before the calculation of Weibull parameters etc. Work is in progress to build a new system based on NetCDF files and Python coding. Until this is ready, it would require a significant effort to recalculate the Weibull parameters for the 6,500+ SAR scenes in this analysis. We have removed this sentence about S-WAsP from the manuscript as it probably leads to confusion rather than clarification:

"S-WAsP is not able to account for seasonal biases in the Weibull parameter estimation. Therefore, no seasonally corrected Weibull parameters or energy densities are available".

Further, we have revised the section on future work to address the issue of sampling biases more clearly using the reviewer's formulation directly:

"With an increasing archive of Sentinel-1 data, future wind atlases will be based on samples, which are more distributed over the time of day. The rapid growth of our SAR data archives over time will in itself improve the accuracy of wind resource statistics. Further, a weighting of the SAR scenes by month could partly overcome seasonal biases and give better estimations of the Weibull parameters while retaining the observational character of a SAR-based wind atlas. Such an advance would be of great help to the SAR wind climatology community and would also impact other meteorological communities which are using the method of moments to fit various distributions to data that is unevenly distributed in space or time."

**MINOR ITEMS**

**Page 1, line 22 - "vary" is vague.** Some readers will read this sentence as meaning the mean wind speed is under 1 m/s rather than the intended meaning of the mean wind speed varying by this much across a wind-farm lease area. This issue of too general terms being used for statistics for which precise terms or phrases are available recurs in this manuscript. I have attempted to point out each location where reader confusion may arise.

**Answer:** We have tried to make this specific sentence more clear to the reader and changed it to:

"Areas designated for offshore wind development by the Bureau of Ocean Energy Management are investigated in more detail; the wind resource in terms of the mean wind speed show spatial variations within each designated area between 0.3 and 0.5 m/s for SAR and less than 0.2 m/s for the WTK."

Thank you for spotting this general weakness; we have tried to focus on precision while revising the manuscript.

**Page 2, line 16** - "at scales around" - This wording will make most readers think the resolution rather than the swath width is several hundred kilometers.

**Answer:** Agreed. We have clarified this while avoiding too satellite specific terms, as the audience is considered to have more of a wind energy background. Changed to. "Scatterometers and synthetic aperture radar (SAR) on board satellites provide coverage over several hundred kilometers and it is possible to retrieve wind speeds at 10 m above sea level from radar backscatter of the ocean surface."

**Page 3, line 1** - "variation" is too vague a term. Please specify if you mean temporal or spatial variation and over what time or space scale.

**Answer:** We have clarified that this is the spatial variation and that scales are approximately a kilometer. The sentence has been changed to:

"Lastly, the spatial variation of mean wind speeds on the kilometre scale are investigated for BOEM lease areas designated for wind farm development."

**Page 4, Table 1** - I suspect most readers would like a column with SAR pixel size. Also, incidence angle and swath width need units. Degrees and Kilometers, I suspect.

**Answer:** We have added the units, thank you for spotting this. This paper is aimed for the audience of wind energy researchers and industry. We do not think that the pixel size is relevant for them as the data presented is averaged to 500m pixels before performing the wind retrieval.

**Page 5, Section 2.4** - It is not clear from this paragraph how these pieces fit together. In particular, it should be made clear whether or not WRF part of WTK?

**Answer:** We have revised section 4.2 and also parts of the abstract and introduction with respect to WTK in order to make it clear that WRF is the model used to create the WTK data set. We now use the abbreviation 'WTK' consistently each time we talk about this data set (previously, we also used the naming 'WIND Toolkit' and occasionally 'WRF').

**Page 5, lines 19-21** - Please explain why the data source switched.

**Answer:** We have added an explanation:

"The switch in wind direction input is present in the database of derived SAR wind maps due to a change to near real time processing."

We now introduce the paragraph by stating the use of a pre-existing data base:

"SAR wind retrievals from the database of the Technical University of Denmark are used for this study and their processing is described in the following."

**Page 6, lines 7-8** - "from modeled wind speeds" - It would help readers to know which modeling system you're referring to here.

**Answer:** It is the same modelled winds as used for the wind retrieval algorithm. We have added this sentence:

"NRCS are calculated from the modelled winds that are used for the SAR wind inversion described in Section 2.4 and compared to the SAR measurements"

**Page 8, lines 6-8** - What are these numbers and why are they being discussed here. Are they extreme cases? Means? The discussion is to too terse for clarity.

**Answer:** We have expanded the description order to point out that the overall bias may be close to zero but there are positive and negative biased for specific wind speed intervals for Envisat in particular:

"The two largest data sets, Envisat (b) and Sentinel-1A AC (e), show a higher mean wind speed from SAR when the buoy wind speed is less than 7 m/s and vice versa lower mean wind speeds from SAR when buoy wind speeds exceed 9 m/s. For Envisat, these opposing biases are averaged to nearly zero in the overall bias".

**Page 10,** paragraph below the second equation - Would it be better to aggregate spatially before fitting the Weibull distribution rather than after? One worries about the order of fitting and smoothing when the fitting is a nonlinear process as it is in this second order moment approach. This is an issue of Jensen's Inequality, I think.

**Answer:** This is in fact done in our analysis. We have moved the relevant description so it is now above the equation to clarify that the spatial mean is taken before the Weibull fit:

"SAR wind images are projected on a regular WGS84 grid with 0.02° cell spacing before processing the data to a wind atlas."

**Page 13, line 12** - While the difference is small in the mean, that is in all likelihood because stable cases and unstable cases are roughly equally likely. The stability impact on the tails of the distribution could thus be quite large. The spatial distribution of biases noted by the authors speak strongly to the impact of surface layer stability on the errors in neutral-equivalent SAR-derived winds, even in the mean.

**Answer:** We agree with this point and have tried to include it in the discussion and future work sections - see our answer to 'Opportunity 1' above.